# Analysis of Volatile Constituents in *Curcuma* Species, *viz. C. aeruginosa*, *C. zedoaria*, and *C. longa*, from Nepal

**DOI:** 10.3390/plants11151932

**Published:** 2022-07-26

**Authors:** Darbin Kumar Poudel, Pawan Kumar Ojha, Anil Rokaya, Rakesh Satyal, Prabodh Satyal, William N. Setzer

**Affiliations:** 1Analytica Research Center, Kritipur, Kathmandu 44600, Nepal; darkwine51@gmail.com (D.K.P.); pojha@aromaticplant.org (P.K.O.); arokaya@aromaticplant.org (A.R.); rsatyal@aromaticplant.org (R.S.); 2Aromatic Plant Research Center, 230 N 1200 E, Suite 100, Lehi, UT 84043, USA; wsetzer@chemistry.uah.edu; 3Department of Chemistry, University of Alabama in Huntsville, Huntsville, AL 35899, USA

**Keywords:** α-turmerone, β-turmerone, *ar*-turmerone, curzerenone, enantiomeric distribution

## Abstract

The genus *Curcuma*, composed of 93 species mainly originating from Asia, Australia, and South America, has been used for medicinal purposes, aromatic, and nutritional values as well as cosmetic. It plays a vital role in flavoring and coloring as well as exhibiting therapeutic agents against different diseases. Nepalese farmers are unaware of the essential oil compositions of *Curcuma* species, *viz. C. aeruginosa*, *C. zedoaria*, and *C. longa*. The investigation of these three essential oils provides insight into their potential as cash crops and earns a reasonable return from their production. The essential oils were obtained from the rhizomes of each plant by hydrodistillation and subjected to Gas Chromatography/Mass Spectrometry (GC–MS) analysis to identify its volatile chemical constituents as well as chiral GC-MS to identify the enantiomeric distribution of chiral terpenoids. The order of extraction yields were *C. longa* (0.89%) > *C. zedoaria* (0.74%) > *C. aeruginosa* (0.37%). In total, the presence of 65, 98, and 84 compounds were identified in *C. longa*, *C. zedoaria*, and *C. aeruginosa*, representing 95.82%, 81.55%, and 92.59% of the total oil, respectively. The most abundant compounds in *C. longa* essential oils were *ar*-turmerone (25.5%), α-turmerone (24.4%), β-turmerone (14.0%), terpinolene (7.2%), β-sesquiphellandrene (5.1%), α-zingiberene (4.8%), β-caryophyllene (2.9%), *ar*-curcumene (1.6%) and 1,8-cineole (1.3%). The most dominant compounds in *C. zedoaria* were curzerenone (21.5%), 1,8-cineole (19.6%), curzerene (6.2%), *trans*-β-Elemene (5.1%), camphor (2.6%), and germacrone (2.3%). The major components in *C. aeruginosa* were curzerenone (59.6%), germacrone (5.3%), curzerene (4.7%), camphor (3.6%), *trans*-β-Elemene (2.6%), and β-eudesmol (1.6%). *C. zedoaria*, and *C. aeruginosa* essential oil from Nepal for the very first time. This study reports for the first time chiral terpenoids from *C. aeruginosa*, *C. zedoaria*, and *C. longa* essential oil. A chemical blueprint of these essential oils could also be used as a tool for identification and quality assessment.

## 1. Introduction

The genus *Curcuma* (Zingiberaceae) is composed of 93 species that mainly originate from Asia, Australia, and South America, and are now cultivated worldwide. Members of this genus have a long history for their medicinal purposes and nutritional values as well as in cosmetics industries [1,2,3]. The major pharmacologically active constituents of *Curcuma* species are curcuminoids and essential oils. Curcuminoids, a mixture of three phenolic compounds namely curcumin, demethoxycurcumin, and bis-demethoxycurcumin, have been proven to posses significant health benefits along with the potential to prevent various diseases [4,5,6]. Essential oils of *Curcuma* species are relatively complex with hundreds of components including terpenes and oxygenated terpenoids. Fragrances and characteristic aromas of *Curcuma* essential oils are due to either a large number of monocyclic bisabolane derivatives or guaiane type sesquiterpenes or germacranes type sesquiterpenes [7]. Different *Curcuma* species essential oils were used in food applications (flavoring, coloring, and preservatives) [8], personal goods (cosmetics and perfumes) [9], and to cope and deal with a variety of ailments (inflammatory conditions of various organs, digestive tract problems, neurodegenerative diseases, wound healing, cancer, viral diseases, and diabetes) [10,11,12,13,14].

The medicinal and culinary properties of the *Curcuma* species rhizome are well-known. Among them, *Curcuma longa*, and *Curcuma zedoaria* were investigated widely around the world due to their high commercial value [15]. On other hand, *Curcuma aeruginosa* essential oil data are less common in the scientific world. *C. longa* is known as yellow turmeric and is most common in the world as a culinary item to enhance flavoring and coloring as well as a good source of antioxidant agents [3]. *C. longa* essential oil had demonstrated good therapeutic properties in diabetes [16], inflammation [17], neuroprotective [18], antimicrobial [19], antioxidant [20] and so on. Most representative components in its essential oil are α-turmerone, *ar*-turmerone, β-turmerone, β-sesquiphellandrene, α-zingiberene, 1,8-cineole, germacrone, *ar*-curcumene, and α-phellandrene [8,21]. *C. zedoaria* rhizome is also called white turmeric and its essential oil is generally composed of 1,8-cineole, curzerenone, camphor, β-caryophyllene, α-elemol, germacrone, curzerene, and β-elemene [22]. The essential oil showed promising activity against cancer [23], diabetes [24], anti-angiogenic [25], and antioxidant [26]. *C. aeruginosa* is known as blue curcuma and its essential oil dominated by curzerenone, 1,8-cineole, camphor, β-pinene, *iso*-borneol, germacrone, curzerene, and curcumenol [15]. The essential oil demonstrated good antimicrobial activity [27], and help in prevention of hair loss [9]. The major compounds of these species are presented in Figure 1. The demand for these three *Curcuma* species essential oils is steadily increasing as a result of a plethora of scientific articles on the health benefits. However, differences in genotype, edaphic variables, pedo-climatic conditions, harvest time, extraction procedure, maturity of rhizome, and analytical procedures all contributed to variations in the chemical composition of *Curcuma* species essential oils [10].

*Curcuma* species can be grown in a variety of soil types as well as in warm and humid climates. Nepal has an extreme altitudinal range with heterogeneous topography with distinct climatic variation and is favorable for the growth of *C. longa*, *C. zedoaria*, and *C. aeruginosa* rhizome. The essential oil of these species has huge demand in the international market. However, Nepalese farmers are still struggling to earn a reasonable return from their production due to lack of sophisticated processing unit and still unknown about the essential oil composition. Hence, there is a dire need to investigate the chemical compositions of rhizome essential oil among *Curcuma* species, *viz. C. aeruginosa*, *C. zedoaria*, and *C. longa*. This is the first research paper from Nepal and includes an in-depth analysis of the *Curcuma* essential oil compositions. Furthermore, the chemical profiles of *Curcuma* species essential oils from Nepal could provide additional chemical fingerprints for identification and quality assessment.

## 2. Results and Discussion

### 2.1. Isolation of Essential Oil and Yields

The fresh mature mother rhizomes of *C. zedoaria*, *C. longa*, and *C. aeruginosa* with light yellow, yellow and blue flesh respectively, were hydrodistilled creating colorless essential oil. The highest extraction yield was observed in *C. longa* (0.89%) followed by *C. zedoaria* (0.74%) and *C. aeruginosa* (0.37%). The extraction yield was quite low, but in close agreement with hydrodistillation yields previously reported [1,22]. In the case of *C. longa*, there were differences in extraction yields; cured had the highest yield, followed by fresh, and then dry rhizome [28]. To the best of our knowledge, maximum yield on hydrodistillation of *C. longa*, *C. zedoaria*, and *C. aeruginosa* had been reported upto 5.5% [29], 1.6% [25], and 0.63% [30], respectively. Genetic variation, harvesting time, the extraction process, and rhizome maturity are all factors that influence extraction yield.

### 2.2. Chemical Composition of Essential Oils

The chemical composition of *C. longa* rhizome essential oils is shown in Table 1. The total number of identified compounds in *C. longa* essential oil is 65 and accounted for 95.82%. The dominant volatile constituents in the *C. longa* essential oil was *ar*-turmerone (25.5%), α-turmerone (24.4%), β-turmerone (14.0%), terpinolene (7.2%), β-sesquiphellandrene (5.1%), α-zingiberene (4.8%), β-caryophyllene (2.9%), ar-curcumene (1.6%) and 1,8-cineole (1.3%), and this is comparable to *C. longa* essential oil compositions grown in North Alabama and India. Furthermore, there was little variation observed between dry and fresh as well as lateral and mother rhizomes [21,28]. Prior research on the essential oils of *C. longa* rhizomes from various geographic locations identified four distinct clusters based on the relative concentrations of turmerones. The first class of clusters is dominated by turmerones, but with relatively large concentrations of other constituents as well such as β-sesquiphellandrene, α-zingiberene, *ar*-curcumene, and 1,8-cineole. The second class of clusters is dominated by turmerones, especially *ar*-turmerone, the third class is dominated by turmerones, especially α-turmerone, and the fourth is dominated by very high concentrations of *ar*-turmerone [15]. So, *C. longa* rhizome essential oils from Nepal fall under the category of the first cluster, i.e., turmerones were dominant, but there were other significant constituents as well. Interestingly, an essential oil from Brazil had *ar*-turmerone, α-zingiberene, β-sesquiphellandrene, humulene epoxide II, *cis*-α-*trans*-bergamotol, and β-turmerone [31]. A sample from India was dominated by 1,8-cineole, β-caryophyllene, α-turmerone, β-turmerone, *ar*-turmerone, and α-phellandrene [32]. All of this suggests that turmerone concentrations, whether high or low, should be desirable in *C. longa* essential oil.

The chemical composition of *C. aeruginosa*, and *C. zedoaria* compositions are presented in Table 2. In the case of *C. zedoaria* essential oil, the total number of identified compounds was 98 and accounted for 81.55%. The most representative compounds werecurzerenone (21.5%), 1,8-cineole (19.6%), curzerene (6.2%), *trans*-β-elemene (5.1%), camphor (2.6%), and germacrone (2.3%). Previous research on the essential oils of *C. zedoaria* rhizomes from India and Nepal revealed two distinct clusters; the first cluster rich in curzerenone/epi-curzerenone followed by camphor, germacrone, 1,8-cineole, and α-copaene and second cluster represented by a single sample with a high concentration of 1,8-cineole [15]. So, *C. zedoaria* essential oils from Nepal fall under the curzerenone/epi-curzerenone chemotype.

The total identified compounds in *C. aeruginosa* essential oil were 86 and accounted for 92.59%. The dominant compounds were curzerenone (59.6%), germacrone (5.3%), curzerene (4.7%), camphor (3.6%), *trans*-β-elemene (2.6%), and β-eudesmol (1.6%). There are only few data on *C. aeruginosa* essential oil. Previous research on the essential oils of *C. aeruginosa* rhizomes from Malaysia, Thailand, and India revealed three distinct clusters. The first cluster is represented by a camphor/germacrone-rich cluster with large concentrations of *iso*-borneol, curzerene, and germacrone. The second cluster was a curcumenol/β-pinene rich cluster. The third cluster was a curzerenone/1,8-cineole cluster [15]. *C. aeruginosa* essential oil from Nepal fall into the curzerenone/1,8-cineole rich chemotype.

Essential oils from the *curcuma* species exhibit impressive biological activities. However, the variations in volatile constituents depending on the geographical location may/may not have same biological activity. *Curcuma* essential oil displayed remarkable antioxidant activity that may be used to minimize the food spoilage in industry. Reactive oxygen species (ROS) in our body initiates the cascade of reaction that leads to different diseases. Antioxidants secondary metabolites remove ROS in body to terminate the oxidative response by free radical [35]. *Curcuma* species essential oil is rich in antioxidant secondary metabolites. *C. longa* essential oil rich in turmerone is thought to be responsible for inhibiting brain-edema formation, inhibiting the key enzymes of diabetes. Besides these *C. longa* essential oil acts as anti-inflammatory, anticancer, antibacterial, and so on. *C. zedoaria* essential oil showed cytotoxicity against different cell line, antidiabetic, antimicrobial as well as larvicidal activity. *C. aeruginosa* essential oil acts as antibacterial, hair re-growth stimulant, and anti-inflammatory. So, these essential oil has been used to treat life threatening diseases, minimizing the food spoilage, cosmetics’, as well as in aromatherapy [10].

### 2.3. Enantiomeric Composition of Essential Oils

In total, 9, 10, and 15 chiral compounds were identified in *C. longa*, *C. zedoaria*, and *C. aeruginosa*, respectively. Relative percentages of the levorotatory (–) and dextrorotatory (+) compounds of *Curcuma* species essential oil are listed in Table 3. The majority of chiral compounds in *C. aeruginosa* were levorotatory. In *C. aeruginosa* essential oil chiral terpenoids such as camphene, β-pinene, linalool, camphor, borneol, and germacrene D exist in dextrorotatory form. On the other hand, sabinene, limonene, bornyl acetate, terpinen-4-ol, β-elemene, and β-caryophyllene exist only in the levorotatory state. In *C. zedoaria* essential oil camphene, linalool, camphor, and germacrene D exists in dextrorotatory form and other detected chiral terpenoids in levorotatory form. Additionally, bornyl acetate, β-caryophyllene, and β-elemene exist in pure levorotatory form. However, germacrene D exists in only dextrorotatory form. α-Pinene, α-phellandrene, *δ-*3-carene, β-phellandrene, α-terpineol, and β-bisabolene dominated in the dextrorotatory form in *C. longa* essential oil. α-Phellandrene, and *δ-*3-carene in absolute dextrorotatory form whereas, (*E*)-nerolidol and β-caryophyllene in absolute levorotatory form. Interestingly, fourteen chiral compounds had been detected from Vietnamese *C. longa* cultivated in North Alabama. However, α-terpineol and α-pinene show contrasting types of enantiomeric distribution as compared to Nepalese essential oil [1]. On the other hand, β-phellandrene and (*E*)-nerolidol were reported for the very first time from Nepalese *C. longa* essential oil. To the best of our knowledge, we have reported chiral terpenoids from *C. longa*, *C. zedoaria*, and *C. aeruginosa* essential oil from Nepal for the very first time which may be a blueprint for identification and authentication.

## 3. Materials and Methods

### 3.1. Plant Material and Isolation of Essential Oils

The Fresh cultivated, mature mother rhizomes of *Curcuma* species (*C. aeruginosa*, *C. zedoaria*, and *C. longa*) were collected in March, 2020 from Kirtipur (27°40’28.8” N 85°15’48.1” E), an elevation of 1348 m. The plants were identified by taxonomists from Central Department of Botany, Tribhuvan University, Kirtipur. The fresh rhizome of each sample (300 g) was cleaned with tap water, and cut into small pieces and was extracted by hydrodistillation in Clevenger apparatus as previously described [36]. The extracted essential oil was dried with anhydrous sodium sulfate and was stored in bottles under 4 °C until use for further studies.

### 3.2. Chemical Composition Analysis by Gas Chromatography/Mass Spectrometry (GC-MS)

Analysis of the chemical constituents in the *Curcuma* species (*C. aeruginosa*, *C. zedoaria*, and *C. longa*) essential oils was carried out using Shimadzu GCMS-QP2010 Ultra under the following condition: mass selective detector (MSD), operated in the EI mode (electron energy = 70 eV), with scan range = 40–400 m/z, and scan rate = 3.0 scans/s. The GC column was a ZB-5MS fused silica capillary with a (5% phenyl)-polydimethylsiloxane stationary phase, a film thickness of 0.25 μm, a length of 60 m, and an internal diameter of 0.25 mm. The carrier gas was helium with a column head pressure of 552 kPa and a flow rate of 1.37 mL/min. The injector temperature was 260 °C, and the detector temperature was 280 °C. The column temperature was set at 50 °C for 2 min and then increased at 2 °C/min to the temperature of 260 °C. For each essential oil sample, 1:10 *v/v* solution in dichloromethane (DCM) was prepared, and 0.3 μL was injected using a split ratio of 1:30. Identification of the individual components of the essential oils was determined by comparison of the retention indices determine by reference to a homologous series of n-alkanes and comparison of the mass spectral fragmentation patterns (over 80% similarity match) with those reported in the literature [33] and our own in-house library [34] using the LabSolutions GC-MS solution software version 4.45 (Shimadzu Scientific Instruments, Columbia, MD, USA). The individual components of *C. longa* essential oil are presented in Table 1 and Table 2 for *C. aeruginosa* and *C. zedoaria* essential oil.

### 3.3. Enantiomeric Analysis by Chiral Gas Chromatography-Mass Spectrometry (CGC-MS)

Chiral GC-MS was carried out as previously reported [37]. Shimadzu GCMS-QP2010S with EI mode (70 eV) and B-Dex 325 chiral capillary GC column was used to perform enantiomeric analysis of *Curcuma* species (*C. aeruginosa*, *C. zedoaria*, and *C. longa*) essential oil. Scans in the 40–400 *m/z* range at a scan rate of 3.0 scan/s. The column temperature was set at 50 °C, at first increased by 1.5 °C/min to 120 °C and then 2 °C/min to 200 °C. The final temperature of the column was 200 °C and was kept constant. The carrier gas was helium with a constant flow rate of 1.8 mL/min. For each essential oil sample, 3% *w/v* solution in DCM was prepared, and 0.1 μL was injected using a split ratio of 1:45. The enantiomer percentages were determined from the peak areas. A comparison of retention times and mass spectral fragmentation patterns with authentic samples obtained from Sigma-Aldrich (Milwaukee, WI, USA) was used to identify the enantiomers. Table 3 shows the enantiomeric distribution of chiral terpenoids from *C. aeruginosa*, *C. zedoaria*, and *C. longa* essential oils.

## 4. Conclusions

In this study, the volatile chemical composition and enantiomeric distribution of *C. aeruginosa*, *C.zedoaria*, and *C. longa* essential oil analyzed by GC-MS and by chiral GC-MS respectively, reported for the first time from Nepal. The extraction yield and chemical composition were comparable to those growing in tropical and subtropical regions of the world, suggesting that these varieties are suitable for commercialization in the international market. These plants volatile constituents’ are not representative of the entire Nepal. Nepal has an extreme altitudinal range with heterogeneous topography with distinct climatic variation that contributes to variations in the chemical compositions of essential oil. However, farmers should be eyeing on systematic production and harvesting as well as investment in sophisticated processing units to create contaminants essential oil as well as increase the quality of *Curcuma* essential oil. The research will surely be useful to encourage farmers to earn a reasonable return from *C. aeruginosa*, *C. zedoaria*, and *C. longa* rhizome production. Additionally, the results of this study can be utilized to provide a baseline for quality assessments of these *Curcuma* species.

## Figures and Tables

**Figure 1 plants-11-01932-f001:**
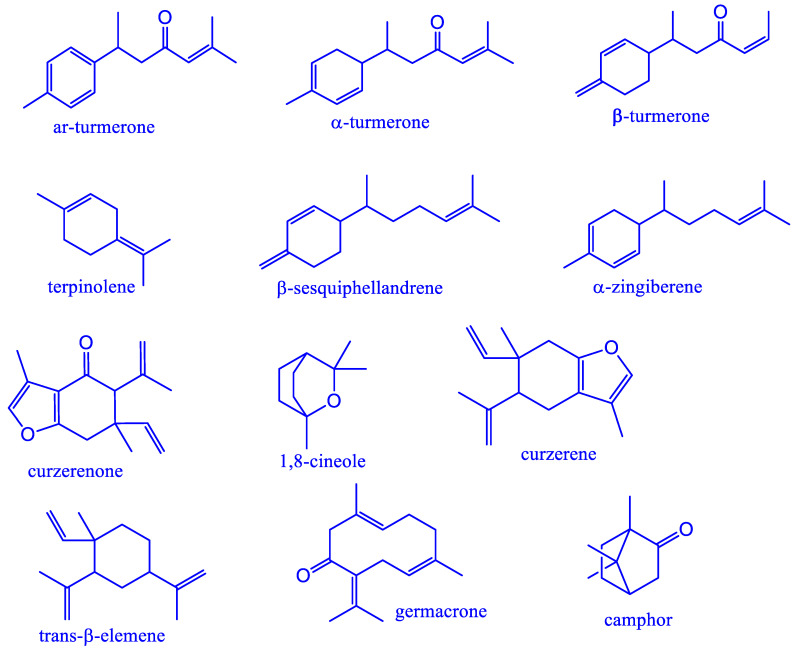
Major compounds present in *C. longa*, *C. zedoaria*, and *C. aeruginosa*.

**Table 1 plants-11-01932-t001:** Individual constituents of *Curcuma longa* essential oil.

RI_calc_	RI_db_	Compound Name	Area% of Constituents in *Curcuma longa*
925	924	α-Thujene	t
933	931	α-Pinene	0.1
972	972	Sabinene	t
978	978	β-Pinene	t
989	989	Myrcene	0.1
1001	1000	δ-2-Carene	t
1007	1008	α-Phellandrene	0.2
1009	1009	δ-3-Carene	0.1
1017	1016	α-Terpinene	0.4
1020	1024	*p*-Cymene	0.1
1024	1028	Limonene	0.1
1025	1029	β-Phellandrene	t
1033	1031	1,8-Cineole	1.3
1035	1034	(*Z*)-β-Ocimene	t
1045	1044	(*E*)-β-Ocimene	t
1057	1058	γ-Terpinene	t
1069	1071	*cis*-Sabinene hydrate	t
1085	1086	Terpinolene	7.2
1090	1091	*p*-Cymenene	0.1
1108	1111	1,3,8-*p*-Menthatriene	t
1130	1132	1(7),3,8-*o*-Menthatriene	t
1138	1143	Epoxyterpinolene	t
1146	1145	Myrcenone	t
1180	1180	Terpinen-4-ol	t
1182	1184	Anethofuran	t
1187	1187	*p*-Cymen-8-ol	0.2
1195	1195	α-Terpineol	t
1234	1237	Turmeric dione	t
1291	1292	2-Undecanone	t
1365	1368	(*Z*,*Z*)-Megastigma-4,6,8-triene	t
1390	1389	7-*epi*-Sesquithujene	t
1402	1405	Sesquithujene	t
1412	1413	*cis*-α-Bergamotene	t
1418	1418	β-Caryophyllene	2.9
1451	1452	(*E*)-β-Farnesene	0.4
1455	1454	α-Humulene	0.5
1476	1464	β-Acoradiene	t
1479	1477	γ-Curcumene	t
1480	1483	*ar*-Curcumene	1.6
1484	1485	*trans*-β-Bergamotene	t
1496	1495	α-Zingiberene	4.8
1508	1507	β-Bisabolene	0.7
1514	1509	β-Curcumene	0.1
1522	1525	β-Sesquiphellandrene	5.1
1528	1528	(*E*)-γ-Bisabolene	0.2
1542	1547	*cis*-Sesquisabinene hydrate	0.2
1560	1561	(*E*)-Nerolidol	0.5
1578	1575	*ar*-Turmerol	0.3
1582	1578	Caryophyllene oxide	0.2
1583	1580	*trans*-Sesquisabinene hydrate	0.3
1595	1599	2,4,4,6-Tetramethyl-6-phenyl-1-heptene	0.3
1611	1612	Humulene epoxide II	0.2
1615	1615	Zingiberenol	0.7
1629	1629	7-*epi-cis*-Sesquisabinene hydrate	0.6
1630	1632	α-Tumerone	0.1
1634	1632	Biotol isomer	0.9
1659	1655	β-Eudesmol	0.1
1664	1662	α-Turmerone	24.4
1667	1664	β-Turmerone	14.0
1668	1665	*ar*-Turmerone	25.5
1681	1684	7-*epi-trans*-Sesquisabinene hydrate	0.3
1688	1685	α-Bisabolol	t
1714	1711	Curcuphenol	0.2
1743	1741	6*S*,7*R*-Bisabolone	0.8
1771	1771	trans-α-Atlantone	0.3

RI_calc_ = Retention index determined with respect to homologous series of n-alkanes on a ZB-5 column. RI_db_ = Retention index from the database [33,34]. ‘t’ indicate trace (< 0.05%).

**Table 2 plants-11-01932-t002:** Individual constituents of *Curcuma aeruginosa* and *Curcuma zedoaria* essential oil.

RI_calc_	RI_db_	Compound Name	Area % of Constituents in	
			*Curcuma zedoaria*	*Curcuma aeruginosa*
882	885	3-Hepten-6-ol	0.1	-
889	888	2-Heptanone	t	-
894	894	2-Heptanol	0.3	0.2
918	921	Tricyclene	t	t
925	924	α-Thujene	t	t
933	931	α-Pinene	0.8	0.2
945	945	α-Fenchene	t	-
949	948	Camphene	1.2	0.8
953	954	Thuja-2,4(10)-diene	t	-
972	972	Sabinene	0.2	t
978	978	β-Pinene	1.7	1.3
989	989	Myrcene	0.3	0.1
998	994	2-Octanol	t	t
1007	1008	α-Phellandrene	t	t
1017	1016	α-Terpinene	t	t
1020	1024	*p*-Cymene	t	t
1024	1028	Limonene	1.1	0.2
1033	1031	1,8-Cineole	19.6	1.2
1035	1034	(*Z*)-β-Ocimene	t	-
1045	1044	(*E*)-β-Ocimene	t	-
1057	1058	γ-Terpinene	0.1	t
1069	1071	*cis*-Sabinene hydrate	t	-
1085	1086	Terpinolene	t	t
1087	1090	2-Nonanone	0.6	t
1090	1090	(3*Z*)-Hexenyl methyl carbonate	t	-
1099	1099	Linalool	0.2	0.3
1105	1103	2-Nonanol	0.6	0.4
1112	1118	*trans*-Thujone	-	t
1119	1122	*trans-p*-Mentha-2,8-dien-1-ol	t	-
1122	1124	*cis-p*-Menth-2-en-1-ol	t	-
1133	1137	*cis-p*-Mentha-2,8-dien-1-ol	t	-
1135	1140	*trans*-Pinocarveol	0.1	-
1141	1145	Camphor	2.59	3.6
1159	1156	*trans*-β-Terpineol	t	t
1164	1161	*iso*-Borneol	-	1.2
1165	1167	*exo*-Acetoxy camphene	t	-
1167	1168	*epi*-Borneol	0.7	-
1170	1170	δ-Terpineol	0.1	-
1173	1171	Borneol	1.2	0.2
1175	1174	*cis*-Pinocamphone	-	t
1180	1180	Terpinen-4-ol	0.4	0.1
1187	1187	*p*-Cymen-8-ol	t	-
1191	1190	2-Decanone	t	t
1195	1195	α-Terpineol	1.1	0.1
1195	1197	*cis*-Piperitol	t	-
1198	1198	2-Decanol	t	-
1206	1206	Verbenone	t	t
1215	1218	*trans*-Carveol	t	-
1226	1232	*cis*-Carveol	0.1	t
1243	1242	Carvone	-	t
1258	1252	*trans*-Myrtanol	t	-
1274	1274	Cyclooctyl acetate	t	-
1283	1282	Bornyl acetate	0.1	t
1287	1288	*iso*-Bornyl acetate	1.7	t
1291	1292	2-Undecanone	0.3	t
1298	1294	*trans*-Pinocarvyl acetate	t	-
1302	1299	Perillyl alcohol	t	-
1310	1307	2-Undecanol	t	t
1331	1333	Bicycloelemene	t	t
1334	1334	δ-Elemene	0.7	0.3
1346	1346	α-Terpinyl acetate	t	-
1381	1381	*cis*-β-Elemene	0.3	0.1
1390	1391	*trans*-β-Elemene	5.1	2.6
1393	1394	*trans*-Sativene	t	t
1418	1418	β-Caryophyllene	0.6	0.7
1428	1430	γ-Elemene	0.3	0.2
1440	1443	6,9-Guaiadiene	t	t
1447	1443	*iso*-Germacrene D	0.2	-
1451	1451	*trans*-Muurola-3,5-diene	-	0.1
1451	1452	(*E*)-β-Farnesene	t	-
1455	1454	α-Humulene	0.1	0.1
1475	1473	*trans*-Cadina-1(6),4-diene	t	t
1476	1475	Selina-4,11-diene	0.1	t
1476	1478	γ-Gurjunene	-	t
1478	1479	α-Amorphene	-	t
1480	1484	Germacrene D	1.9	0.8
1483	1487	Guaia-1(10),11-diene	t	-
1496	1488	δ-Selinene	t	t
1488	1489	β-Selinene	0.5	0.3
1494	1493	Curzerene	6.2	4.7
1496	1495	Aciphyllene	0.1	-
1498	1498	α-Selinene	t	0.2
1504	1502	*trans*-β-Guaiene	0.2	0.1
1504	1503	β-Dihydroagarofuran	-	0.1
1508	1505	Germacrene A	0.1	t
1513	1514	γ-Cadinene	-	t
1517	1516	Cubebol	-	t
1518	1519	δ-Cadinene	0.1	0.1
1540	1540	Occidentalol	t	t
1549	1549	α-Elemol	0.1	0.1
1558	1560	Germacrene B	0.7	0.5
1560	1561	(*E*)-Nerolidol	t	-
1582	1578	Caryophyllene oxide	0.1	0.2
1590	1584	Globulol	0.7	0.4
1595	1592	Viridiflorol	-	0.4
1601	1594	(*E*)-β-Elemenone	0.3	0.5
1605	1600	Curzerenone	21.5	59.6
1607	1607	5-*epi*-7-*epi*-α-Eudesmol	-	0.2
1610	1610	*iso*-Curzerenone	-	0.6
1614	1616	Curcumenol	1.3	-
1622	1623	10-*epi*-γ-Eudesmol	-	0.2
1630	1627	*iso*-Spathulenol	0.6	0.7
1630	1631	γ-Eudesmol	t	0.5
1646	1637	Agarospirol	-	0.2
1646	1640	*epi*-α-Cadinol	0.1	0.2
1659	1655	β-Eudesmol	-	1.6
1652	1657	α-Eudesmol	0.5	-
1658	1659	Selin-11-en-4α-ol	0.6	0.4
1660	1660	Selin-11-en-4β-ol	0.1	-
1670	1663	Bulnesol	-	0.2
1693	1692	Germacrone	2.3	5.3
1705	1708	Aromadendrane-4,10-diol A	-	t
1710	1708	Thujopsenal	-	0.1
1777	1775	Curzerenone A	-	0.5
1790	1795	Curzerenone B	-	t
1828	1832	*iso*-Germacrone C	t	t
1843	1834	Curcumenone	0.5	-
1989	1992	4-Methoxystilbene	t	-
2366	2376	Butyl stearate	t	-

RI_calc_ = Retention index determined with respect to homologous series of n-alkanes on a ZB-5 column. RI_db_ = Retention index from the database [33,34]. ‘t’ indicate trace (< 0.05%).

**Table 3 plants-11-01932-t003:** Enantiomeric distributions of chiral terpenoids of *Curcuma aeruginosa*, *Curcuma zedoaria*, and *Curcuma longa* essential oil.

Chiral Terpenoid Compound	*Curcuma aeruginosa*	*Curcuma zedoaria*	*Curcuma longa*
α-Pinene	(+)24.7: (–)75.3	(+)43.5: (–)56.5	(+)70.6: (–)29.4
Camphene	(+)90.7: (–)9.3	(+)92.3: (–)7.7	-
β-Pinene	(+)59.1: (–)40.9	(+)47.8: (–)52.2	-
Sabinene	(+)0: (–)100	-	-
α-Phellandrene	-	-	(+)100: (–)0
*δ*-3-Carene	-	-	(+)100: (–)0
Limonene	(+)0: (–)100	-	(+)45.6: (–)54.4
β-Phellandrene	-	-	(+)91.4: (–)8.6
Linalool	(+)62.4: (–)37.6	(+)55.3: (–)44.7	-
Camphor	(+)99.8: (–)0.2	(+)92.3: (–)7.7	-
Bornyl acetate	(+)0: (–)100	(+)0: (–)100	-
Terpinen-4-ol	(+)0: (–)100	-	-
*δ*-Elemene	(+)45.0: (–)55.0	(+)33.7: (–)66.3	-
α-Terpineol	(+)29.1: (–)70.9		(+)51.5: (–)48.5
Borneol	(+)100: (–)0		-
β-Elemene	(+)0: (–)100	(+)0: (–)100	-
β-Caryophyllene	(+)0: (–)100	(+)0: (–)100	(+)0: (–)100
Germacrene D	(+)100: (–)0	(+)100: (–)0	-
β-Bisabolene	-	-	(+)92.9: (–)7.1
(*E*)-Nerolidol	-	-	(+)0: (–)100

## Data Availability

All data are available in the article.

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
