# Peer review of "Analysis of Volatile Constituents in Curcuma Species, viz. C. aeruginosa, C. zedoaria, and C. longa, from Nepal"

_plants, 2022, doi:10.3390/plants11151932_

Round 1

Reviewer 1 Report

In table 1 heading trace can add as a footnote of table 

In table 1 Area% of Curcuma longa can be modified 

In table 2 trace can add as footnote of table 

structure of major compounds identified can add in results 

minor english language mistakes can be modified

Author Response

  1. In table 1 heading trace can add as a footnote of table

Response: It has been corrected as per your suggestion.

  1. In table 1 Area% of Curcuma longa can be modified 

Response: It has been corrected as per your suggestion.

  1. In table 2 trace can add as footnote of table

Response: It has been corrected as per your suggestion.

  1. Structure of major compounds identified can add in results

Response: The structure of major compounds present in curcuma species has been added in the results and discussion section.

  1. Minor english language mistakes can be modified

Response:  The manuscript has been proofread by a native speaker of English, and the errors are corrected as per your suggestion.

Reviewer 2 Report

The authors aimed to characterize essential oils extracted from nepalese Curcuma species. The authors say that their results might be helpful for authenticating and identifying these essential oils. To do this, it is not enough to characterize just one plant sample for each species. You have to increase the number of samples to provide representative fingerprints of nepalese curcuma essential oils. Moreover, you characterize the essential oils obtained from one specific region, so the chemical profile might not be representative of entire Nepal considering that pedoclimatic conditions usually have a strong impact on the composition of essential oils.

For these reasons, I do not suggest for publication in Plants. 

Author Response

The authors aimed to characterize essential oils extracted from Nepalese Curcuma species. The authors say that their results might be helpful for authenticating and identifying these essential oils. To do this, it is not enough to characterize just one plant sample for each species. You have to increase the number of samples to provide representative fingerprints of Nepalese curcuma essential oils. Moreover, you characterize the essential oils obtained from one specific region, so the chemical profile might not be representative of entire Nepal considering that pedoclimatic conditions usually have a strong impact on the composition of essential oils.

For these reasons, I do not suggest for publication in Plants.

Response: We have collected the each sample (300 g) from Kirtipur, Nepal.  To the best of our knowledge this is the first paper on curcuma species Kirtipur, Nepal and characterized essential oil and compared it with other nation essential oil from previous published data. This research provides the data to compared for adulteration detection, and provide a baseline for quality assessments in near future. There was little data available on C. aeruginosaC. zedoaria essential oil. We clearly mentioned that there is no variation in chemical composition of essential oil if there was no any variation in genetics, edaphic variables, climate, harvest time, extraction procedure, and maturity of rhizome.

Reviewer 3 Report

The article entitled ”Analysis of Volatile Constituents in Curcuma Species, viz. C. aeruginosaC. zedoaria, and C. longa, from Nepal” describes the GCMS analysis of volatile oils from three different curcuma species from Nepal, in addition to their chiral GC analysis. The article is well written and organized and can be published after the following corrections:

1-    The authors reported that the oil obtained from C. aeruginosa is violet colored; is there any reason for this color from the volatile oil composition (as it is extremely similar to that of zedoaria which is coloreless). Did the authors tried extracting the oil using other method such as solvent extraction to detect if the color is produced in all cases or it is resulting from decomposition of certain components by hydro distillation similar to the famous case of chamomile.

2-    As a suggestion, is the difference between C. longa and the other two species has an impact on their use and /or biological effect. The authors can add a paragraph discussing this point to enrich the article discussion.

3-    Introduction line 43: have proven to possess significant health benefits instead of (be) 

4-    Line 48: different curcuma species essential oils were

5-    Results line 102: dominant volatile ; please add space

Author Response

The article entitled ”Analysis of Volatile Constituents in Curcuma Species, viz. C. aeruginosaC. zedoaria, and C. longa, from Nepal” describes the GCMS analysis of volatile oils from three different curcuma species from Nepal, in addition to their chiral GC analysis. The article is well written and organized and can be published after the following corrections:

1. The authors reported that the oil obtained from C. aeruginosa is violet colored; is there any reason for this color from the volatile oil composition (as it is extremely similar to that of zedoaria which is coloreless). Did the authors tried extracting the oil using other method such as solvent extraction to detect if the color is produced in all cases or it is resulting from decomposition of certain components by hydro distillation similar to the famous case of chamomile.

Response: Thank you for proving such mistake in our manuscript. Clarification has been added. Actually, flesh of C. aeruginosa rhizomes was blue.

2. As a suggestion, is the difference between C. longa and the other two species has an impact on their use and /or biological effect. The authors can add a paragraph discussing this point to enrich the article discussion.

Response: It has been added as per your suggestion in results and discussion section.

3. Introduction line 43: have proven to possess significant health benefits instead of (be) 

Response: As per your suggestion, it has been corrected

4. Line 48: different curcuma species essential oils were

Response: Corrected

5. Results line 102: dominant volatile; please add space

Response: Corrected

Reviewer 4 Report

Criticism

1. The harvest phase of the raw material, lateral and / or mother rhizomes, as well as its pre-processing (purification, degree of grinding, etc.) to obtain essential oil is not indicated.

2. It is not indicated by whom the identification of the species was carried out?

3. Harvest raw materials from wild and cultivated plants?

4. All results are presented in one repetition, which raises the question of their reliability. Are such studies sufficient to draw fundamental conclusions?

5. It is not indicated how the selection of raw materials for the study was carried out, i.e. were the rhizomes of individual plants analyzed, or were plants sampled in a population, or something else?

6. It is not indicated how the results were processed, whether hierarchical cluster analysis was performed, what the authors include in the concept of “chemotype” and how it differs from the concept of “cluster”. How enantiomeric analisys of chiral terpenoids of these essential oils may be used for identification and quality assessment, how does the composition of chiral terpenoids change during plant growth and development?

Author Response

  1. The harvest phase of the raw material, lateral and / or mother rhizomes, as well as its pre-processing (purification, degree of grinding, etc.) to obtain essential oil is not indicated.

Response: Clarification has been added to section 3.1.

  1. It is not indicated by whom the identification of the species was carried out?

Response:  Clarification has been added to section 3.1.

  1. Harvest raw materials from wild and cultivated plants?

Response: All the plants taken in our research were cultivated.

  1. All results are presented in one repetition, which raises the question of their reliability. Are such studies sufficient to draw fundamental conclusions?

Response: Yes, we has compared our research essential oil of curcuma to other different nation and found that the essential oil fall on either cluster. There was so little data available to compared in case of C. aeruginosaC. zedoaria. We have only studied the chemical composition by GC-MS and chiral GC-MS. This indicates that our samples are potent in commercialization in international market.

  1. It is not indicated how the selection of raw materials for the study was carried out, i.e. were the rhizomes of individual plants analyzed, or were plants sampled in a population, or something else?

Response: The rhizomes of individual plants analyzed and we try to explore the essential oil composition. Nepal is categorized as low economy country and Nepalese are struggling to earn a reasonable return from their production. If we, Nepalese aware the chemical composition of essential oil that may improve economy. Besides these, the demand of essential oil increase drastically every day due to these essential oil deals with different life threatening disease.

  1. It is not indicated how the results were processed, whether hierarchical cluster analysis was performed, what the authors include in the concept of “chemotype” and how it differs from the concept of “cluster”. How enantiomeric analysis of chiral terpenoids of these essential oils may be used for identification and quality assessment, how does the composition of chiral terpenoids change during plant growth and development?

Response: The results of our study compared with previously published data, based on major compounds, where cluster analysis was done and differentiate the different chemotypes. We used the mature mother rhizomes, the enantiomeric distribution of chiral terpenoids were almost same if there was no any variation in genetics, edaphic variables, climate, harvest time, extraction procedure, maturity of rhizome. Enantiomeric distribution of chiral terpenoids may be used in adulteration detection. Based on the chiral ratio we can categorize the essential oil either pure or adulterate.

Round 2

Reviewer 2 Report

The authors should highlight in the discussion or conclusions that the plants were collected from a specific region and that the compositional results are not representative of the entire Nepal. Pedoclimatic conditions are important factors to be considered along with other factors. The authors should discuss this aspect either in the discussion or in the conclusions. 

Here some examples:

doi.org/10.3390/molecules26206157

doi.org/10.21577/0103-5053.20210146 

Author Response

Response: Clarification has been added in the conclusions section. Thanks for your valuable response. 

Reviewer 4 Report

The corrections made by the authors are clear and sufficient.

Author Response

Response: Thank you very much for your compliments, and highly appreciate it.